# Assessment of a Postharvest Treatment with Pyrimethanil via Thermo-Nebulization in Controlling Storage Rots of Apples

**Felix Büchele** [1,*], **Daniel A. Neuwald** [1], **Christian Scheer** [1], **Rachael M. Wood** [1], **Ralf T. Vögele** [2] **and Jens N. Wünsche** [3]

[1] Competence Centre for Fruit Growing at Lake Constance (KOB), 88213 Ravensburg, Germany; neuwald@kob-bavendorf.de (D.A.N.); scheer@kob-bavendorf.de (C.S.); rachael.wood@kob-bavendorf.de (R.M.W.)

[2] Institute of Phytomedicine, Phytopathology University of Hohenheim, 70599 Stuttgart, Germany; ralf.voegele@uni-hohenheim.de

[3] Institute Crop Science, Crop Physiology of Specialty Crops, University of Hohenheim, 70599 Stuttgart, Germany; jnwuensche@uni-hohenheim.de

[*] Correspondence: felix.buechele@kob-bavendorf.de

**Abstract:** Apples are very susceptible to infections from various fungal pathogens during the growing season due to prolonged exposure to environmental influences in the field. Therefore, a strict and targeted fungicide strategy is essential to protect fruit and trees. Increased environmental and health concerns and pathogen resistance have resulted in a rising demand to reduce fungicide usage and residues on marketed fruit. Thus, producers must develop new plant protection strategies to conform to the legal and social demands while still offering high-quality apples. This study assessed the efficacy of a post-harvest fungicide treatment with pyrimethanil via thermo-nebulization for controlling storage rots and compared the results to those of standard pre-harvest fungicide strategies. The results showed that a single post-harvest application of pyrimethanil successfully controlled storage rots and is comparable to strategies using multiple pre-harvest fungicide applications. The control of fungal rot was sustained even after 5 months of storage and 2 weeks of shelf life. Thermo-nebulization into the storage facility allowed for a lower dosage of fungicide to be used compared to pre-harvest applications, while still maintaining optimal rot control. Residue analyses showed that the post-harvest fungicide treatment did not exceed legal or retailer's standards.

**Keywords:** fungicide; fungal pathogens; residues; resistance; pollution; *neofabraea* spp.

## 1. Introduction

Apples provide favorable conditions for fungal growth due to their rich availability of nutrients, high water content and suitable pH. Consequently, storage rot is the predominant cause of production losses in integrated apple production worldwide [1]. Throughout the growing season, fruit and trees are susceptible to infections by several fungal pathogens. At harvest, fruit can appear free from infection despite an infection occurring during the season, long before harvest. These fruit can then develop symptoms during storage, packing, transit, or marketing [2]. Due to this long timespan, apple production requires the frequent application of chemical or organic fungicides. Fungal decay poses a serious concern for producers, retailers, and consumers. Insufficient protection due to limited fungicide application can not only result in visually flawed fruit and crop losses but also cause lasting damages to the tree material [3]. Furthermore, some fungal species such as *Penicillium expansum* or *Alternaria alternata* produce significant amounts of mycotoxins during their growth, posing serious health risks to humans after consumption [4,5]. The mycotoxin patulin could be detected in the fruit tissue of healthy-looking pears inoculated with *P. expansum*, demonstrating the importance of a strict fungicide strategy to prevent infections [6]. While it may be essential in apple production, the extensive use of fungicides during pre-harvest stages comes with serious drawbacks. Large volumes of fungicides are

required due to losses during application and degradation in the field. Both synthetic and organic fungicides can accumulate in soil and groundwater, which can have detrimental effects on the ecosystem and many organisms [7]. Plant protection in horticulture is especially criticized due to the side effects on beneficial organisms such as honeybees. Prior studies report that fungicides can affect honeybee food consumption, metabolism, and immune response [8].

Therefore, during the registration process, applications must verify that the product used for plant protection does not have any adverse effects on the survival and development of bee colonies.

The frequent use of fungicides has led to rising cases of pathogens acquiring resistance and, subsequently, a reduced efficacy in controlling storage rot [9]. Consequently, even more fungicide applications are required to produce healthy apples. Characteristics of both pathogen and fungicide influence the development of resistant fungal populations. For example, *P. expansum* and *Botrytis cinerea* can rapidly acquire fungicide resistance due to their short life cycles, high genetic variability, and abundant reproduction [10]. The risk of fungicide resistance is classified by the Fungicide Resistance Action Committee [11] based on the target site and mode of action of fungicides and on whether the treatment is chemical or biological.

The primary concern of consumers refers to the supposed health risks posed by the consumption of fruit with pesticide residues [12]. In the EU, products are tested regularly, and maximum residue levels (MRL) are defined for every fruit and fungicide combination. The MRLs are set below an acute reference dose for which no negative impacts after consumption can be expected. Fruit produced within the EU rarely exceed these limits [13], and no detrimental health effects are expected for 99% of the typical fungicide residues [14]. Fungicide residues are necessary to guarantee protection against rots during storage and marketing. Therefore, lowering the residue levels would yield no additional health benefit for consumers, as the chance of an infection and possible accumulation of mycotoxins would increase, thus increasing the health risks to the consumer [14].

The broad-spectrum fungicide pyrimethanil is part of the chemical class of anilinopy-rimidines and was first registered for post-harvest use on pome fruit in 2004 in the USA. It is classified as medium-risk for fungicide resistance, with known cases of resistance regarding *B. cinerea* and *P. expansum* [15,16]. Two modes of action have been proposed for pyrimethanil: (i) the inhibition of the biosynthesis of methionine and other amino acids [17], (ii) the blockage of the secretion of hydrolytic enzymes involved in the infection process of fungal pathogens [18]. Pyrimethanil is highly effective in inhibiting conidial germination and germ tube elongation of *P. expansum* [19] and mycelial growth of *B. cinerea* [20]. The post-harvest application of pyrimethanil via drenching is highly efficient in controlling storage rots in apples [20,21] and citrus fruit [22]. Previously, pyrimethanil was used against apple scab (*Venturia inaequalis*); however, the emergence of resistance resulted in reduced efficacy [23], making additional fungicide treatments close to harvest necessary [24].

Fungal infections usually occur through injuries occurring mostly during harvest or post-harvest handling, hence, after the last fungicide spray. Due to the timing, pre-harvest fungicide sprays are limited in the control of storage rots [25]. Therefore, alternative methods for fungicide application have been developed to limit the detrimental effects on the environment, slow the development of resistance, and improve the timing of application. Post-harvest fungicide treatments via drenching or dripping in a fungicide solution are common practice but are present various challenges. Waste removal can be complicated and expensive, and the efficacy may be reduced by foreign objects in the solution. Furthermore, there is a significant potential of accumulating spores and distributing the inoculum from one bin to another [26]. Reusing contaminated bins without proper sanitation in the following season may also promote the development of resistant pathogens [15]. For a single producer, investing in the required technology may also be expensive and not practical.

The method of applying fungicides directly into the storage room via thermo-nebulization was proposed in 2000 [27]. Fine fungicide particles are generated by an aerosol electrical

generator at a high temperature ($\pm190\ ^\circ$C) and distributed by a forced airflow through a window into the storage facility. Factors such as stacking pattern, bin design, and air circulation need to be considered before application to achieve a uniform distribution [28]. Insufficient attention to these factors may result in uneven distribution and subsequently insufficient protection or increased residue levels. Several studies describe the promising efficacy of fungicide application via hot fogging without exceeding maximum residue levels [29–31].

The aim of this study was to (1) investigate the potential of a post-harvest pyrimethanil application via thermo-nebulization to control storage rots in apple; (2) compare a post-harvest harvest pyrimethanil application to regular pre-harvest fungicide strategies; (3) measure residue levels on fruit to determine their marketability.

## 2. Materials and Methods

### 2.1. Experimental Design

The study was conducted with the apple cultivar 'Pinova' at the Competence Centre for Fruit Growing at Lake Constance (KOB) in Southern Germany. Trees were planted in 2011 on M9 rootstocks at a spacing of $3.2 \times 1.0$ m and, at the time of the experiment, were 2.0 m tall. All trees received identical fungicide treatments prior to the experiment to guarantee homogeneity (Table 1).

**Table 1.** Fungicide application with respective dosage per hectare (ha) prior to the experiment. Identical for all treatments.

| Date | Active Ingredient | Dosage per ha | Date | Active Ingredient | Dosage per ha |
|---|---|---|---|---|---|
| 23 March | Copper hydroxide | 0.6 L | 05 June | Dithianon/Myclobutanil | 0.25 kg + 0.125 L |
| 01 April | Dithianon/Sulfur | 0.25 kg + 2 kg | 13 June | Captan/Myclobutanil | 1 L + 0.125 L |
| 08 April | Dodin | 0.625 L | 18 June | Dithianon/Penconazol | 0.25 kg + 0.125 L |
| 25 April | Fluxapyroxad/Dithianon | 0.1 L + 0.25 kg | 01 July | Dithianon | 0.25 kg |
| 27 April | Dodin | 0.625 L | 11 July | Captan | 0.6 kg |
| 01 May | Fluxapyroxad/Captan | 0.1 L + 0.625 kg | 19 July | Captan | 1 L |
| 07 May | Captan/Trifloxystrobin | 0.625 kg | 05 August | Dithianon | 0.25 kg |
| 17 May | Dithianon/Sulfur | 0.25 kg + 2 kg | 08 August | Captan | 0.6 kg |
| 22 May | Calciumpolysulfid | 6 L | 15 August | Dithianon | 0.25 kg |
| 27 May | Dithianon/Sulfur | 0.25 kg + 2 kg | 27 August | Captan | 0.75 kg |

By September, the month before harvest, the experimental site was divided into seven treatments areas plus an untreated control (1) in a randomized block design. Treatments 2–8 had various combinations of fungicides with different modes of action, applied either pre- or post-harvest. Each treatment contained four replicates of 11 trees (Table 2).

Apples were harvested on 4 October 2019 and moved to the storage facility. In separate storage rooms, Treatments 2, 4, and 6 received a post-harvest pyrimethanil treatment via thermo-nebulization. The product used was Xedathane-HN (156 g pyrimethanil/L) at an application rate of 0.05 L/t apples. Treatments 1, 3, 5, 7, and 8 received no additional fungicide treatment after harvest. Apples were stored for 5 months at 3 $^\circ$C in regular atmosphere (RA). After treatment application and storage, samples were taken from each treatment and evaluated for fungicide residue levels in an external laboratory, excluding the control (Treatment 1).

**Table 2.** Fungicide treatments, dosage, and time of application.

| | | Days to Harvest | | | | Post |
|---|---|---|---|---|---|---|
| | | −28 | −21 | −14 | −7 | +3 |
| **Treatments** | 1 | | | | | |
| | 2 | | | | | Pyrimethanil |
| | 3 | Captan | Captan | | | |
| | 4 | Captan | Captan | | | Pyrimethanil |
| | 5 | Captan | Captan | Trifloxystrobin | Trifloxystrobin | |
| | 6 | Captan | Captan | Trifloxystrobin | Trifloxystrobin | Pyrimethanil |
| | 7 | Captan | Captan | Pyrimethanil | Pyrimethanil | |
| | 8 | Captan | Captan | Fludioxonil | Fludioxonil | |
| **Dosage** | Captan<br>Trifloxystrobin<br>Pyrimethanil<br>Fludioxonil | 0.75 kg/ha<br>0.05 kg/ha<br>0.375 L/ha<br>0.15 kg/ha | | 0.05 L/t | | |

## 2.2. Evaluations

After storage, fruit were visually examined for the presence of storage rot. Each treatment contained four replicates of 200 apples. Apples were classified as either healthy or, based on the rot symptoms, infected with *Neofabraea* spp., *Monilinia fructigena*, *Botrytis cinerea*, *Fusarium* spp., *Alternaria* spp., *Neonectria ditissima*, or *Penicillium* spp. For each replicate, the ratio of infected to total fruit was recorded, and the statistical mean of the whole experimental treatment was calculated. Infected fruit were discarded, and fruit showing no symptoms were stored for another two weeks at 20 °C ± 1 °C in regular atmosphere to simulate shelf life. Following shelf life, a second evaluation was performed following the same procedure.

## 2.3. Residue Level Analysis

For the analysis of fungicide residues, 10 fruit per treatment were taken (i) immediately following the post-harvest pyrimethanil treatment and (ii) after 5 months of storage. The analysis of plant protection residues on fruit was carried out by Labor Friedle GmbH using the QuEChERS approach [32]. An initial single-phase extraction of 10 g of the sample was performed with 10 mL of acetonitrile. By adding 4 g of anhydrous $MgSO_4$ and 1 g NaCl, liquid–liquid partitioning was obtained. Simultaneously, the removal of residual water and clean-up were performed by dispersive solid-phase extraction. For this, 150 mg of anhydrous $MgSO_4$ and 25 mg of primary secondary amine sorbent were mixed with 1 mL of acetonitrile extract. The quantitative analysis of plant protection residues was carried out using gas chromatography/mass spectrometry.

## 2.4. Quality Analysis

At harvest, fruit maturity was determined according to the ripening index proposed by Streif [33] with analysis of starch degradation, fruit firmness, and total soluble solids. Additionally, titratable acidity was measured. Quality analysis was carried out following the standard procedure [34]. Eight fruits per replicate were examined. To determine starch degradation, fruit were cut in half and brushed with Lugol's iodine. The progress was rated on a scale from 1 to 10 (1 representing fruit with no starch degradation, and 10 representing fruit with completely hydrolyzed starch). Fruit firmness was tested in the equatorial region of the fruit, using a Güss Fruit Texture Analyzer penetrometer with an 11 mm tip, between the sun and the shadow side, after removing the epidermis. The total soluble solids (TSS) content was analyzed by refractometry using a 1 mL sample of juice. Total titratable acidity was determined by titration with 0.1 N NaOH of a 10 mL fruit juice

solution diluted in 50 mL of distilled water. After RA storage and 2 weeks of shelf life, the quality tests were repeated.

### 2.5. Statistical Analysis

This study was conducted with a completely randomized design. Analysis of variance (ANOVA) was carried out to identify differences between means of treatments ($p < 0.05$). Data that showed significant difference were subjected to the Tukey's test ($p < 0.05$). In figures, means with no significant differences are displayed with identical letters. Statistical analysis was performed using R version 4.1.1.

### 3. Results

#### 3.1. First Evaluation

After storage, the untreated control had the highest percentage of fruit infested with storage rot. When a single post-harvest treatment of pyrimethanil (Treatment 2) was applied, the proportion of infestation was significantly reduced from 13.8 % in the untreated control to 2.6 % (Figure 1A). Treatment 3, which consisted of two captan applications applied 28 and 21 days before harvest, had the second highest percentage of storage rot and did not differ significantly from the control group. When pyrimethanil was applied post-harvest to the same treatment (Treatment 4), storage rot was reduced. The combination of pre-harvest captan and trifloxystrobin produced a lower incidence of storage rot compared to the control group (Treatment 5). The additional post-harvest treatment of pyrimethanil (Treatment 6) did not yield any further reductions in storage rot. Similar results to those of Treatments 5 and 6 were achieved with the combination of pre-harvest captan and pyrimethanil (Treatment 7), as well as of captan and fludioxonil (Treatment 8). There was no difference in storage rot between the pre-harvest applications of pyrimethanil (Treatment 7), trifloxystrobin (Treatment 5), and fludioxonil (Treatment 8) when applied together with captan.

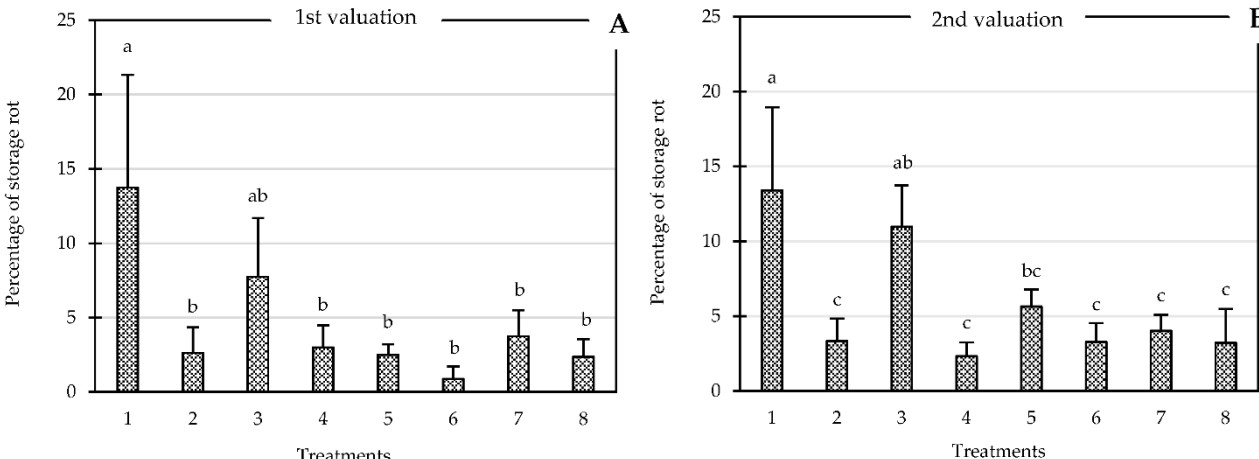

**Figure 1.** Storage rot percentage as affected by various pre- and post-harvest fungicides after (**A**) 5 months of storage and after (**B**) additional 2 weeks of shelf life. Bars with identical letters do not differ significantly (Tukey Test = 0.05). The standard deviation of the mean is denoted by a capped bar at the top of the column.

#### 3.2. Second Evaluation

Fruit that showed rot symptoms in the first evaluation were discarded. The second evaluation was conducted after fruit were kept for 2 weeks at 20 °C ± 1 °C and RA to simulate shelf life conditions. The control group displayed the highest incidence of storage rot (Figure 1B), and a single post-harvest pyrimethanil treatment significantly reduced storage rot infestation (Treatment 2).

Overall, improved efficacy in controlling storage rots compared to the control was observed for Treatments 2, 4, 6, 7, and 8, with no significant differences among these treatments. Treatment 3 did not differ significantly from the control group. Although a lower infestation than in the control group was achieved by combining captan and trifloxystrobin pre-harvest (Treatment 5), captan alone (Treatment 3) did not significantly reduce the infection rates compared to the control. Fungal rot incidence did not differ between evaluation times for each treatment ($p > 0.05$).

### 3.3. Identification of Individual Rots

Between 60 to 90% of the total infestation could be attributed to bull's eye rot caused by *Neofabraea* spp. (Table 3). Fruit in the control group and Treatment 3 (two pre-harvest captan applications) had a significantly higher percentage of bull's eye rot than the other treatments. Significantly lower bull's eye rot incidences were observed in fruit that received a post-harvest pyrimethanil treatment (Treatments 2, 4, and 6), regardless of pre-harvest fungicide strategies. Even though the pre-harvest combination of captan and trifloxystrobin (Treatment 5) showed a lower efficacy in controlling bull's eye rots than Treatments 2, 4, 6, and 8, fruit receiving Treatment 5 had a lower infestation rate than the control group. Bull's eye rot incidences were also reduced with the pre-harvest combinations of captan and fludioxonil (Treatment 8) or captan and pyrimethanil (Treatment 7), although the outcome of the latter strategy did not differ from that of Treatment 5.

**Table 3.** Percentage of total rot incidences and of the responsible fungal pathogens in both evaluations with standard deviation of the mean. Means with identical letters do not differ significantly (Tukey Test $\alpha = 0.05$). Abbreviations: Tr.: Treatments; neof.: *neofabraea* spp.; n.d.: *Neonectria ditissima*; m.f.: *monilia fructigena*; p.e.: *penicillium* spp.; fr.: *fusarium* spp.; b.c.: *botrytis cinerea*; a.a.: *Alternaria* spp.

| Tr. | Total | neof. | n.d. | m.f. | p.e. | fr. | b.c. | a.a. |
|---|---|---|---|---|---|---|---|---|
| 1 | 25.0 ± 11.1 a | 22.8 ± 11.9 a | 0.5 ± 0.0 a | 0.3 ± 0.5 a | 1.1 ± 0.9 a | 0.1 ± 0.3 a | 0.0 ± 0.0 a | 0.3 ± 0.5 a |
| 2 | 5.9 ± 2.5 c | 3.8 ± 1.6 c | 0.5 ± 0.4 a | 0.8 ± 1.5 a | 0.5 ± 0.4 a | 0.3 ± 0.3 a | 0.1 ± 0.3 a | 0.0 ± 0.0 a |
| 3 | 17.9 ± 4.3 ab | 15.4 ± 4.7 ab | 0.4 ± 0.5 a | 0.3 ± 0.5 a | 1.1 ± 0.9 a | 0.0 ± 0.0 a | 0.1 ± 0.3 a | 0.6 ± 0.8 a |
| 4 | 5.3 ± 2.3 c | 3.3 ± 2.2 c | 0.9 ± 0.5 a | 0.9 ± 0.3 a | 0.3 ± 0.5 a | 0.0 ± 0.0 a | 0.0 ± 0.0 a | 0.0 ± 0.0 a |
| 5 | 8.0 ± 1.6 bc | 6.5 ± 1.1 bc | 0.1 ± 0.3 a | 0.0 ± 0.0 a | 0.8 ± 0.6 a | 0.4 ± 0.5 a | 0.0 ± 0.0 a | 0.3 ± 0.3 a |
| 6 | 4.1 ± 1.4 c | 2.6 ± 1.1 c | 0.9 ± 1.0 a | 0.5 ± 0.6 a | 0.1 ± 0.3 a | 0.0 ± 0.0 a | 0.0 ± 0.0 a | 0.0 ± 0.0 a |
| 7 | 7.6 ± 1.0 bc | 6.0 ± 0.7 bc | 0.5 ± 0.6 a | 0.0 ± 0.0 a | 0.6 ± 0.5 a | 0.3 ± 0.5 a | 0.0 ± 0.0 a | 0.1 ± 0.3 a |
| 8 | 5.5 ± 3.2 c | 4.0 ± 2.7 c | 0.6 ± 0.6 a | 0.5 ± 0.4 a | 0.3 ± 0.5 a | 0.0 ± 0.0 a | 0.0 ± 0.0 a | 0.1 ± 0.3 a |

### 3.4. Residue Analysis

Residue analysis was performed after post-harvest fungicide treatment and after the storage period. Table 4 indicates that none of the fungicide strategies adopted in this study exceeded the legal and retailer's limits. Captan residues, the metabolite tetrahydrophtalamide, and dithianon were measured in all tested treatments (Figure 2). The residue levels of captan increased when it was applied in the month before harvest. In general, when active ingredients were part of the fungicide program, their residues could always be identified in the analysis, even after months of storage. The limits of the maximum number of measurable residues were not exceeded.

The highest captan residue levels were recorded after Treatment 6 at 1.0 mg/kg; however, this value is far below the legal limits of 10.0 mg/kg (Table 4). Significantly higher residues of pyrimethanil were measured when the application was carried out post-harvest. For captan, trifloxystrobin, and dithianon, the residue levels decreased during the 5 months of RA storage. In contrast, pyrimethanil residues were higher in the second analysis, regardless of the application timing. Treatment 4 had the highest pyrimethanil residues after storage at 3.4 mg/kg, which is substantially lower than the corresponding legal maximum level of 15 mg/kg. Treatment 6 showed the highest trifloxystrobin residue level, corresponding to 0.079 mg/kg, thus within the legal limit of 0.7 mg/kg. Fludioxonil residues remained constant during storage, at 0.13 mg/kg.

**Table 4.** Fungicide residues [mg/kg] after harvest (1st evaluation) and after 5 months of storage (2nd evaluation).

| Treatments | | | | | | | | |
|---|---|---|---|---|---|---|---|---|
| **1st evaluation** | **2** | **3** | **4** | **5** | **6** | **7** | **8** | **MRL** [2] |
| Captan [1] | 0.26 | 0.74 | 0.75 | 0.90 | 1.00 | 0.95 | 0.57 | **10.00** |
| Dithianon | 0.04 | 0.02 | 0.05 | 0.06 | 0.04 | 0.08 | 0.03 | **3.00** |
| Pyrimethanil | 2.30 | | 2.70 | | 2.40 | 0.48 | | **15.00** |
| Trifloxystrobin | | | | 0.08 | 0.08 | | | **0.70** |
| Fludioxonil | | | | | | | 0.13 | **5.00** |
| **2nd evaluation** | **2** | **3** | **4** | **5** | **6** | **7** | **8** | **MRL** [2] |
| Captan [1] | 0.08 | 0.55 | 0.51 | 0.59 | 0.61 | 0.53 | 0.49 | **10.00** |
| Dithianon | 0.06 | 0.05 | 0.06 | 0.05 | 0.04 | 0.04 | 0.02 | **3.00** |
| Pyrimethanil | 2.90 | | 3.40 | | 3.00 | 0.65 | | **15.00** |
| Trifloxystrobin | | | | 0.05 | 0.05 | | | **0.70** |
| Fludioxonil | | | | | | | 0.13 | **5.00** |

[1] Captan values include the metabolite tetrahydrophtalamide. [2] Maximum residue levels [mg/kg] as defined by the EU pesticide database.

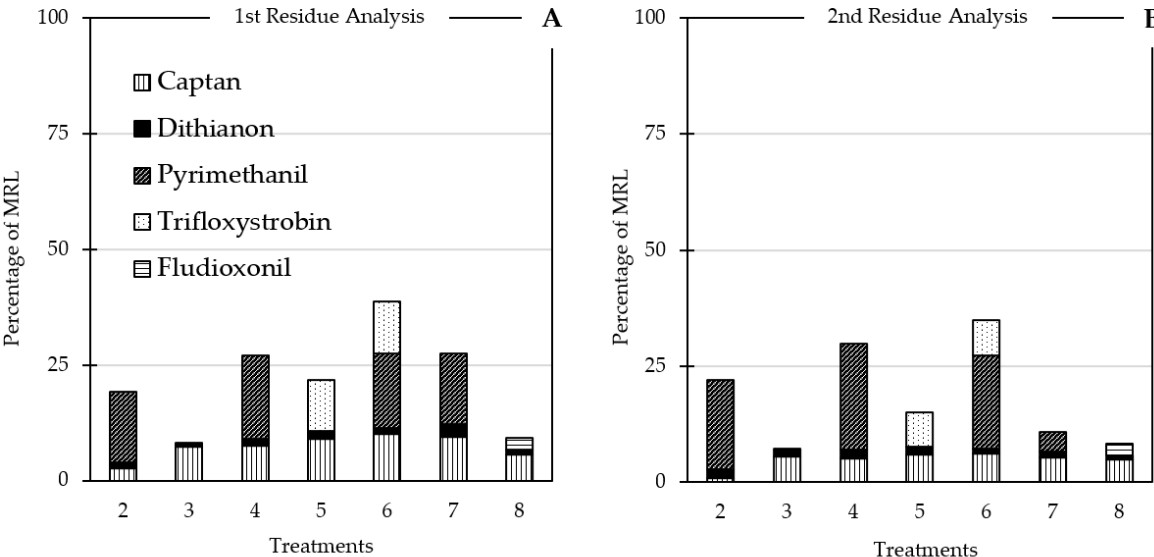

**Figure 2.** Results of MRL analysis of experimental treatments 2 to 8 after the application of pyrimethanil shortly after harvest (**A**) and after 5 months of RA storage (**B**). Cumulated percentages of maximum residues.

*3.5. Quality Analysis*

Following storage and shelf life, firmness decreased in all treatments by an average of about 20 N compared to the harvest value (Figure 3A). Firmness did not significantly differ among the various fungicide strategies. Furthermore, the fungicide strategy, storage, and shelf life did not affect TSS (Figure 3B). For titratable acidity, Treatment 6 had the highest content after storage and shelf-life, while Treatments 2 and 8 had the lowest. In general, titratable acidity was halved during storage and shelf life compared to the value recorded at harvest (Figure 3C).

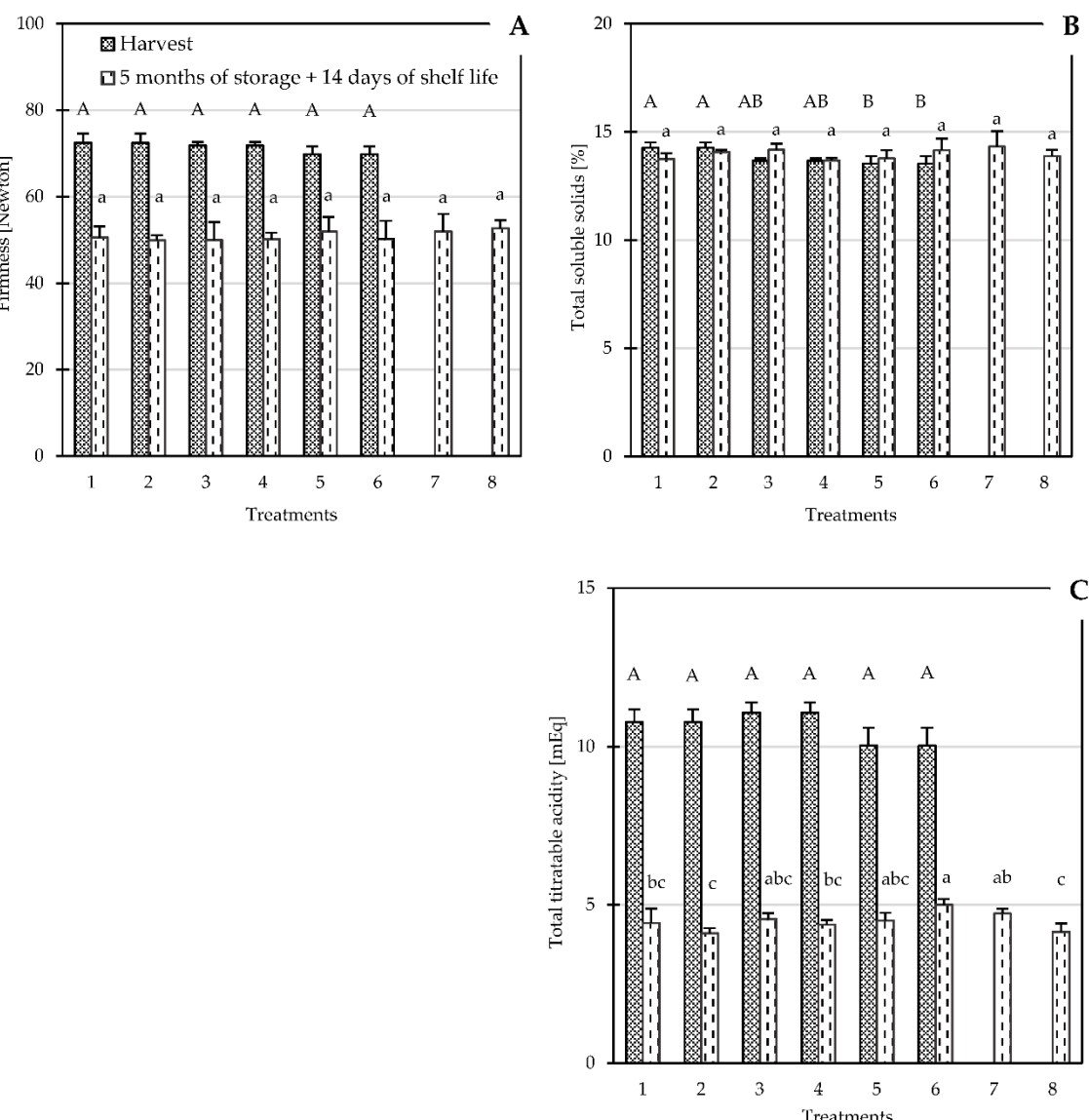

**Figure 3.** (**A**) Fruit firmness, (**B**) total soluble solids content, and (**C**) titratable acidity at harvest and after 5 months of storage plus 2 weeks of shelf life, as affected by various pre- and post-harvest fungicide applications. Means with identical uppercase letters did not differ significantly at harvest, and means with identical lowercase letters did not differ significantly after storage and shelf life (Tukey Test $\alpha = 0.05$). The standard deviation of the mean is denoted by a capped bar at the top of the column.

## 4. Discussion

The high percentage of storage rot in the control group at the first evaluation indicated that the fungicide applications administered before the experiment's start (Table 1) were insufficient to control storage rots (Figure 1). In contrast, a single post-harvest application of pyrimethanil in Treatment 2 significantly reduced the incidence of storage rot. These results are consistent with those of multiple studies that demonstrated a comparable efficacy [31,35].

Combining fungicide applications before and after harvest did not provide any further benefit. A single post-harvest application of pyrimethanil in Treatment 2 provided particularly good protection from storage rot and appeared to be capable of replacing pre-harvest fungicides at least in the month before harvest.

The limited fungicide strategy of two pre-harvest captan treatments (Treatment 3) did not significantly reduce storage rot incidence compared to the control treatment (Figure 1). These results demonstrate the importance of a consistent fungicide strategy until harvest and the challenge encountered when trying to reduce fungicide usage. Due to the potentially high losses, multiple fungicide treatments just before harvest are needed to reduce the risk of post-harvest diseases [36]. Reduced application rates or extended intervals between applications may also contribute to the development of fungal pathogen resistance, lowering the possible efficacy of a reduced fungicide strategy even further [37]. However, the relationship between application rate and resistance risk is not fully established and may vary according to the fungicide used [38].

Adequate protection against fungal pathogens was attained with pre-harvest combinations of captan with trifloxystrobin (Treatment 5), pyrimethanil (Treatment 7), or fludioxonil (Treatment 8) (Figure 1). A good efficacy of the use of the active substances individually has already been demonstrated [36]. Furthermore, studies showed that the post-harvest application of fludioxonil by thermo-nebulization had a moderate efficacy in controlling *P. vagabunda*, a bull's eye rot species [31]. Differences in the effectiveness of pyrimethanil treatments with different application times were not observed after 5 months of storage, confirming previous studies that showed similar efficacies of pyrimethanil when applied pre- or post-harvest [29].

The storage of the remaining healthy-appearing fruit under "shelf conditions" produced a similar pattern to the first evaluation, and infection rates did not differ between evaluations for each treatment. Thus, the effect of the fungicide strategies on storage rot incidence remained constant under "shelf conditions". Consequently, a putative prolonged protective effect by post-harvest treatments due to the direct application on fruit material and the reduced degradation of the active ingredient during storage cannot be shown in this study. It has been argued that when applied post-harvest, pyrimethanil can still provide good protection from *P. expansum* infection months after harvest due to its persistence on apple fruit during storage and marketing [39]. Therefore, additional fungicide treatments during packing would be unnecessary. However, protection against fungal pathogens may decrease, when fruit are stored at room temperature, although this was not observed in our study.

Bull's eye rot caused by *Neofabraea* species is the dominant storage rot in the Lake Constance region of Germany. The cultivar 'Pinova' is particularly susceptible to this fungal pathogen [40]. This was confirmed in our experiment, where bull's eye rot represented the dominant cause of storage rot (Table 3). The highest percentage was recorded in the untreated control and in fruit subjected to Treatment 3. A combination of captan and trifloxystrobin showed an improved performance in controlling bull's eye rot (Treatment 5) compared to the control. Treatment 5 represents a typical fungicide strategy in the Lake Constance Region, for which good efficacy in protecting against *Neofabraea* spp. has already been demonstrated for the cultivar 'Pinova' [41]. Pyrimethanil's notable efficacy in inhibiting *Neofabraea* spp. has previously been demonstrated for pre- and post-harvest treatments [29].

Residue analysis demonstrated that no fungicide strategy exceeded the legal ("EU Pesticides Database") and retailer requirements [36] concerning the maximum level and the number of residues, and hence, all fruit were marketable. These results indicate that the post-harvest treatment of pyrimethanil did not substantially increase residues and the associated health risks. Even if the degradation of fungicides is lower during storage than in the fields due to the lack of abiotic influences, the thermo-nebulization process allows for a lower dosage of fungicide to be used and, thus, avoids exceeding the maximum residue levels. Fruit treated with pyrimethanil before harvest (Treatment 7) displayed lower residue levels after storage and shelf life than fruit subjected to Treatments 2, 4, and 6, which received a post-harvest fungicide application (Table 4). The complete avoidance of any fungicide residues on fruit is neither possible nor advisable. Pyrimethanil residues following a post-harvest treatment can be detected after 7 months of CA storage [39].

However, in this aforementioned study, the treatment was administered via drenching. The authors found that the remaining residues still possessed a fungicidal effect and protected against various storage rots, even after storage. Indeed, our study confirms that pyrimethanil effectively controls rots throughout storage and shelf life.

The residue analysis showed that thermo-nebulization is an efficient method for administering fungicides evenly on the fruit surfaces without exceeding the allowed maximum levels (Figure 2), validating previous studies [28,35,42]. Although no fruit were treated with dithianon in the month before harvest, all fruit tested positive for dithianon residues. Prior to the experiment, all trees received identical fungicide protection (Table 1). The emergence of fungicide resistance of apple scab to anilinopyrimidines has made multiple preventative applications of dithianon and captan during the growing season an essential part of the plant protection strategy in the Lake Constance region [41].

Residue levels persisted throughout storage but did not exceed the legal limits for any treatment. Additionally, captan residues were detected in fruit subjected to Treatment 2, despite receiving no application during the study. However, multiple captan treatments did occur for scab control prior to the experiment. The fact that fungicide residues from applications administered months before harvest are still detectable after 5 months of storage needs to be considered when implementing the plant protection strategy so that legal limits are not exceeded with an additional post-harvest fungicide treatment.

The dominant form of marketing presents the greatest challenge in introducing post-harvest fungicide treatments in the Lake Constance region. Few producers can afford storage units of commercial size, conduct their own marketing, and sell their entire harvest directly to the consumer. Most supply their products to producer groups or organizations. Therefore, apples of the same cultivar but from multiple producers are kept in the same storage room. Before carrying out a post-harvest fungicide application, organizations need to be certain that no fruit will exceed the maximum allowed number of measurable residues. This requires adapting the fungicide strategy for all producers supplying the organizations and an increased organizational effort.

At harvest, apples had a 'Streif ripening index' of about 0.07 (data not shown). According to the Competence Centre's recommendation for the cultivar 'Pinova' in the Lake Constance Region, the optimum harvest window (0.16 to 0.08) was already passed. With later harvest dates, and therefore an advanced maturity, apples become more susceptible to infections by fungal pathogens [43]. Thus, it is possible that in this analysis, higher incidences of storage rot were observed than would normally be expected.

Even though a loss of quality was identified for firmness and titratable acidity, no effect of the fungicide programs on fruit storability was observed. Previous studies on the effects of fungicides mostly focused on the possible detrimental influence on the yield and growth of trees and less on the quality parameters of the fruit [44]. However, fungicide usage can reduce leaf area due to a phytotoxic effect, thus lowering the photosynthesis rate and affecting fruit quality [45].

Some significant differences were observed at harvest for total soluble solids; however, there are insufficient data to identify the fungicide treatment as the cause.

Our study demonstrates that the various fungicide treatments did not affect 'Pinova' apples' quality degradation during storage. However, there was a considerable loss of quality from harvest to the post-storage period, likely due to RA storage. Controlled atmosphere (CA) settings are a commercially well-established strategy to improve quality maintenance compared to regular atmosphere. Furthermore, some studies demonstrated a slower fungal growth in fruit stored under CA [46,47]. A low temperature is reported to be the most important factor in controlling rots in storage [48]; however, atmosphere modification could still offer further benefits when combined with a post-harvest pyrimethanil treatment.

A possible area for future research is combining fungicide application via thermo-nebulization with 1-MCP, an ethylene inhibiter, to reduce quality loss during storage, especially, since there are conflicting reports on the effect of 1-MCP on the development of storage rots [49,50].

## 5. Conclusions

The post-harvest treatment with pyrimethanil maintained a sufficient control of storage rot, equal to that achieved with various pre-harvest fungicide strategies consisting in multiple treatments with different active components. These results indicate that a single post-harvest pyrimethanil treatment can adequately substitute multiple fungicide treatments administered in the month prior to harvest. Residue analysis showed no treatment exceeded the legal or retailer's maximum residue levels or the maximum number of measurable residues. These results demonstrate that a post-harvest fungicide application via thermo-nebulization could be a viable strategy for decreasing storage rot incidence while simultaneously preventing detrimental effects to the environment, reducing the risk of resistance development, and minimizing chemical waste. Only the insufficient protection against storage scab and the possibly increased organization effort in managing the number of residues pose a significant challenge for producers.

**Author Contributions:** Conceptualization, D.A.N. and C.S.; methodology, D.A.N. and C.S.; validation, F.B., D.A.N., C.S., R.M.W., R.T.V. and J.N.W.; formal analysis, F.B.; investigation, F.B., D.A.N. and C.S.; resources, D.A.N. and C.S.; data curation, F.B.; writing—original draft preparation, F.B.; writing—review and editing, F.B., D.A.N., C.S., R.M.W., R.V. and J.N.W.; visualization, F.B.; supervision, D.A.N. and C.S.; project administration, D.A.N. and C.S. All authors have read and agreed to the published version of the manuscript.

**Funding:** This research received no external funding.

**Institutional Review Board Statement:** Not applicable.

**Informed Consent Statement:** Not applicable.

**Data Availability Statement:** The data are contained within the article.

**Conflicts of Interest:** The authors declare no conflict of interest.

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
