# Peer review of "Assessment of a Postharvest Treatment with Pyrimethanil via Thermo-Nebulization in Controlling Storage Rots of Apples"

_agronomy, doi:10.3390/agronomy12010034_

Round 1
Reviewer 1 Report
Studies comparing pre- and postharvest application of fungicides are rare, therefore this study gives important insight into a more integrated approach of postharvest decay management using fungicides before and after harvest. The manuscript is generally well written, just lacks some details on the aspect of the fungal identification i.e. which Neofabraea species were found?
An important aspect that is missing in the presented data is that the results of the statistical analysis of the different amounts of decay reported in each treatment will have to be added to table 3 or be shown in a graph (similar to Figure 3).
The tables (3 and 4) should be formatted differently do the headers and content of each column can be read.
The following minor changes are recommended:
Page 1, Line 30: change 'penicillium exp.' to 'P. expansum'
Page 2, Line 11: change to 'P. expansum or B. cinerea'
Page 2, Line13: change 'side' to 'site'
Page 2, Line 20: remove 'in' after 'within'
Page 4, Line 36: spell out 'Eight fruit' instead of starting the sentence with '8 fruits'
Page 5, Line 6: remove 'Each treatment contained four replicates of 200 apples' (this is part of materials and methods and already mentioned there)
Page 5, Line 23: give specific temperature of 'room temperature' in degree Celsius
Page 5, Line 36: italicise 'Neofabraea'
Table 3: use abbreviation for species names i.e. 'P. e.' for 'P. expansum' to make column header readable and spell out species names in the footnote of the table. Add letters showing statistically significant difference between treatments.
Table 4: round numbers to two digits to be consistent and make table readable.
Figure 2: Format bars in colours such as black, white and grey or use patterns to allow graph to be read when printed in black and white
Reviewer 2 Report
The present study reported a post-harvest fungicide treatment with pyrimethanil via thermo-nebuliza-tion for controlling storage rots, compared to standard pre-harvest fungicide strategies. It provides a potential way to explore the postharvest quality control of apple fruit. The topic is interesting and the manuscript is good in order, and some minor revisions should be cared.
- Please check the tense used in unique forms, for instance in Abstract “Results show that a single post-harvest”…,”showed”; in conclusion, “These results indicate”…”indicated”
- The data expression of art should be improved. Please keep the data in table in unique forms, two numbers after decimal, such as in Table 3 and Table 4. The descriptions in table 3 are not clear, they should be carefully revised and make them clearly. Values in table are not easy to read. Please reorganize them.
Round 2
Reviewer 1 Report
Please italicise all latin species names.
Author Response
(1) Please italicise all latin species names.
All latin species names are now italicized.
Made changes:
p. 6 neofabraea spp.
p. 6. abbreviations in Table 3